# Investigation and Causes of Spontaneous (Non-Diabetic) Hypoglycaemia in Adults: Pitfalls to Avoid

**DOI:** 10.3390/diagnostics13203275

**Published:** 2023-10-22

**Authors:** Maiar Elghobashy, Rousseau Gama, Raashda Ainuddin Sulaiman

**Affiliations:** 1Clinical Chemistry, Black Country Pathology Services, Royal Wolverhampton Trust, Wolverhampton WV10 0QP, UK; m.elghobashy1@nhs.net (M.E.); rousseau.gama@nhs.net (R.G.); 2School of Medicine and Clinical Practice, Wolverhampton University, Wolverhampton WV1 1LY, UK; 3Department of Medical Genomics, Center for Genomic Medicine, King Faisal Specialist Hospital and Research Center, Riyadh 11211, Saudi Arabia; 4College of Medicine, Alfaisal University, Riyadh 11533, Saudi Arabia

**Keywords:** hypoglycaemia, spontaneous hypoglycaemia, neuroglycopaenia, non-diabetic, adult

## Abstract

Although adult spontaneous (non-diabetic) hypoglycaemia is rare, its recognition is important for the preventative or curative treatment of the underlying cause. Establishing Whipple’s triad—low blood glucose, neuroglycopaenia and resolution of neuroglycopaenia on increasing blood glucose levels to normal or above—is essential to verify hypoglycaemia. Awareness that hypoglycaemia may occur in severely ill patients is important for its prevention. Further investigation in such cases is unnecessary unless another cause of hypoglycaemia is suspected. Patients are often asymptomatic and normoglycaemic at review. Their history of medication, self-medication, access to hypoglycaemic drugs, alcohol use and comorbidity may provide aetiological clues. The investigation involves obtaining blood samples during symptoms for laboratory glucose measurement or provoking fasting or postprandial hypoglycaemia as directed by symptoms. If confirmed, insulin, c-peptide, proinsulin and beta-hydroxybutyrate are analysed in hypoglycaemic samples. These will classify hypoglycaemia due to non-ketotic hyperinsulinaemia, non-ketotic hypoinsulinaemia and ketotic hypoinsulinaemia, and direct investigations to identify the underlying cause. There are, however, many pitfalls that may mislabel healthy individuals as “hypoglycaemic” or misdiagnose treatable or preventable causes of spontaneous hypoglycaemia. Clinical acumen and appropriate investigations will mostly identify hypoglycaemia and its cause.

## 1. Introduction

Hypoglycaemia is not a diagnosis but a manifestation of a disease process with many causes. In practice, however, it occurs most frequently in patients with diabetes mellitus, usually due to over-treatment, particularly with insulin. Although rare, spontaneous hypoglycaemia in non-diabetic adults is important to identify as preventative or curative treatment is often available.

Spontaneous hypoglycaemia often poses a diagnostic challenge as patients are mostly healthy-looking individuals who present with non-specific neuroglycopaenic symptoms. It is, therefore, crucial to have a high index of suspicion, confirm the presence of spontaneous hypoglycaemia and evaluate the individual for an underlying cause. This article reviews clinical and biochemical investigations, with an emphasis on common pitfalls to avoid in the identification of spontaneous hypoglycaemia in adults and its underlying causes.

## 2. Pathophysiology of Hypoglycaemia

The cornerstone investigations in elucidating the aetiology of documented hypoglycaemia are the measurement of insulin, c-peptide and proinsulin in hypoglycaemic samples.

The pancreatic secretion of insulin is stimulated by glucose absorption. It returns to basal levels within two to four hours as glucose concentrations fall and glucagon secretion rises on completion of glucose absorption. Insulin and c-peptide, cleavage products of proinsulin, are co-secreted from the pancreas in equimolar concentrations. The shorter half-life and hepatic extraction of insulin ensures that the c-peptide:insulin molar ratios are >1 in the peripheral circulation. A small amount of proinsulin secreted into the portal circulation appears in the peripheral circulation as its hepatic extraction is low [1].

Auto-regulation of pancreatic insulin and glucagon secretion maintains circulating glucose between 3.5 mmol/L and 10 mmol/L in normal health. Homeostatic mechanisms to counteract hypoglycaemia principally involve stimulating the sympathetic nervous system and increased counter-regulatory hormonal (glucagon, catecholamines, growth hormone and cortisol) responses. The overall effect of these is to suppress insulin (c-peptide and proinsulin) secretion, promote hunger, increase glucose output by stimulating glycogenolysis and gluconeogenesis, decrease peripheral tissue glucose uptake and provide alternative fuels for brain function through lipolysis and ketogenesis [2,3,4].

## 3. Verification of Hypoglycaemia 

### 3.1. Biochemical Definition

Hypoglycaemia is defined as a plasma glucose concentration of 3.0 mmol/L or less. Its definition is arbitrary and owes its importance to the fact that hypoglycaemia of this severity produces brain dysfunction and suppresses insulin secretion [3,4,5]. The equivalent blood glucose concentration cut-off for hypoglycaemia is 2.5 mmol/L or less since whole blood glucose is 15% lower than the corresponding plasma glucose [6].

Pitfalls to avoid are failure to recognise spurious hypoglycaemia and pseudohypoglycaemia. Spurious hypoglycaemia is a pre-analytical artefact due to either delayed separation of plasma from blood cells and incomplete inhibition of sample tube glycolysis or excessive in vitro glucose consumption due to leucocytosis, erythrocytosis or thrombocytosis [7,8]. Pseudohypoglycaemia has been used to describe falsely low finger prick capillary glucose values in patients with poor peripheral circulation [9]. Pseudohypoglycaemia has also been used to describe falsely low postprandial glucose in venous samples [10]. In the fasting state, arterial and venous blood glucose levels are almost identical. Following meals, however, as the tissues take up the glucose, the venous plasma glucose may be up to 2.5 mmol/L lower than the corresponding arterial blood sample [10]. Lower venous plasma glucose levels may, therefore, be misinterpreted as hypoglycaemia in the postprandial state. In such cases, free-flowing capillary blood samples should be taken as they more accurately reflect arterial blood glucose concentrations, which determine hypoglycaemic symptoms.

### 3.2. Neuroglycopaenia

Hypoglycaemic symptoms are manifested by neurological dysfunction and, therefore, are termed neuroglycopaenia. The two predominant neuroglycopaenic syndromes are acute and sub-acute neuroglycopaenia [11]. Acute neuroglycopaenia is common in iatrogenic hypoglycaemia and presents with sweating, anxiety, tremor, palpitations, sweating, hunger and paraesthesia (adrenergic symptoms). Sub-acute neuroglycopaenia, as usually observed in spontaneous hypoglycaemia but also as hypoglycaemic unawareness in patients with diabetes, presents with episodic disorientation, behavioural changes, amnesia and loss of consciousness. Clinical features common to acute and sub-acute neuroglycopaenia include transient hemiplegia, strabismus, hypothermia, hyperthermia, convulsions and automatism. Prolonged, untreated, severe hypoglycaemia may cause brain death but is rare because of efficient counter-regulatory mechanisms [11,12,13].

### 3.3. Whipple’s Triad

Neuroglycopaenic symptoms are idiosyncratic but non-specific. Acute and sub-acute neuroglycopaenia may, therefore, only be confidently confirmed when Whipple’s triad is fulfiled, namely, with neuroglycopaenic symptoms, a low blood glucose and symptoms relieved by raising blood glucose to, or above, normal levels [11,14].

## 4. Investigation of Documented Hypoglycaemia

### 4.1. Patients Who Are Ill

Critically ill, hospitalised patients with sepsis, hypothermia and organ (hepatic, renal, cardiac and respiratory) failure may develop spontaneous hypoglycaemia (Table 1). Although COVID-19 infection is well known to cause hyperglycaemia both in diabetic and non-diabetic individuals [15], hypoglycaemia has been reported in COVID-19 patients in those admitted to ICU [16] or who received treatment with hydroxychloroquine or colchicine [17,18]. It, therefore, appears COVID-19 is not a cause of hypoglycaemia per se in the absence of severe comorbidity or medication associated with hypoglycaemia.

It is often sufficient to recognise the association of underlying disease with hypoglycaemia and take preventive action without further investigation. Confirmation of the underlying mechanism may, however, be obtained (Figure 1).

### 4.2. Apparently Well Individuals

Spontaneous hypoglycaemia should be considered following a single episode or recurrent sub-acute neuroglycopaenia; otherwise, diagnosis of the underlying disease may be delayed or missed.

Ideally, a blood sample should be collected when symptomatic to confirm or exclude hypoglycaemia and, if confirmed, a blood sample offers the best prospect of identifying its underlying aetiology [19,20]. A glucose value of 4 mmol/L or higher excludes hypoglycaemia, whereas values between 3.1 to 3.9 mmol/L may require further investigation, depending on the level of clinical suspicion [19].

However, patients are often asymptomatic when medically reviewed; at this time, blood glucose concentrations are invariably non-diagnostic. A careful history should then be reviewed that includes prescribed medication associated with hypoglycaemia (Table 2), over-the-counter supplements and herbal remedies (which may be contaminated with insulin secretagogues [21,22,23,24,25]), dietary toxins (such as unripe ackee nuts and mushrooms causing liver failure) [26], potential access to hypoglycaemic drugs and alcohol intake should be ascertained as these may provide aetiological clues. A history of comorbidity may also be relevant, particularly a history of bariatric or major gastric surgery.

The history should then be directed as to whether symptoms occur during fasting or in a postprandial state to guide further investigations. Investigations focus on obtaining blood samples during symptoms for laboratory glucose measurement or, failing that, by provoking fasting or postprandial hypoglycaemia. If hypoglycaemia is confirmed, appropriate investigations are performed to elucidate the underlying aetiology (Figure 1).

Provocation of a hypoglycaemic attack involves fasting with or without exercise when fasting (post-absorptive) hypoglycaemia is suspected or giving a mixed meal when postprandial (alimentary) hypoglycaemia is suspected. Other provocative tests are of limited value in the initial investigation of hypoglycaemia because of poor diagnostic specificity and sensitivity [1].

Obtaining a blood sample during symptoms involves training the patient, relative or friends to collect a capillary blood sample into a suitable capillary tube or onto specially prepared filter paper for laboratory blood glucose measurement, and if hypoglycaemia is confirmed, further investigation is mandatory [27].

Although invaluable in alerting patients with diabetes to iatrogenic hypoglycaemia, caution should be applied to the use of blood glucose meters to detect spontaneous hypoglycaemia as these may erroneously label healthy individuals as having hypoglycaemia [19,28,29]. Inappropriate use of glucose meters may also potentially misclassify patients with genuine, spontaneous hypoglycaemia as having normoglycaemia [1]. Similarly, continuous interstitial glucose monitoring (CGM) devices used to manage the treatment of patients with diabetes are not recommended for the detection of spontaneous hypoglycaemia except perhaps for those following bariatric surgery [29].

### 4.3. Tests for Provoking Hypoglycaemia

#### 4.3.1. Overnight Fast

Most patients with episodic spontaneous hypoglycaemia will have at least one prolonged overnight fasting (18 h) plasma glucose concentration of <2.5 mmol/L when measured on three separate occasions [30]. Hypoglycaemic episodes may appear asymptomatic unless patients are specifically tested for mild cognitive dysfunction.

#### 4.3.2. Prolonged Fast

The prolonged fast (with ambulation) for up to 72 h is considered the gold standard for investigating fasting hypoglycaemia [31]. It is expensive as it requires hospital admission, medical supervision and close blood test monitoring and, fortunately, is not often required. It is, therefore, reserved for patients in whom hypoglycaemia is strongly suspected but have not had a documented hypoglycaemic episode, either spontaneously or following an overnight fast.

During the fast, the patient is allowed to drink non-caloric, caffeine-free beverages and encouraged to be ambulatory. Venous blood samples are collected every 6 h until plasma glucose falls to 3.5 mmol/L. Samples are then collected every 1–2 h, and the patient is regularly assessed for neuroglycopaenia. Blood samples are immediately analysed for plasma glucose, and serum is frozen for later measurement, if required, of pancreatic B-cell products, insulin antibodies, beta-hydroxybutyrate (β-OHB) and appropriate drug screen. The fast is terminated when plasma glucose falls below 2.5 mmol/L and the patient has neuroglycopaenic symptoms. After the collection of appropriate specimens for further tests, the patient is fed. In the absence of symptoms and hypoglycaemia, the test is terminated at 72 h, and the patient is fed [29].

It has been reported that a 48 h fast is diagnostically as efficient and, therefore, should replace the recommended 72 h fast [32]. Others, however, report that the 72 h fast minimises misdiagnosis and maximises the probability of diagnosing an insulinoma [31].

Rarely, a patient with insulinoma may intermittently have low or suppressed insulin during hypoglycaemia with breakthrough ketosis, but c-peptide and proinsulin concentrations remain inappropriately high [29].

Following prolonged fasting, a few healthy individuals, usually young women, may have plasma glucose concentrations of 3.0 mmol/L or less and may be misdiagnosed as having ketotic hypoinsulinaemic hypoglycaemia. They, however, do not develop symptoms [19,33], emphasising the importance of recognising neuroglycopaenia.

#### 4.3.3. Mixed Meal Test

The mixed meal test is used to investigate patients who experience postprandial symptoms of postprandial (alimentary) hypoglycaemia [19,30]. There is no standard meal, but it should include the components known to cause the symptoms. Ideally, free-flowing capillary blood samples are collected before and at 30 min intervals for five hours after ingesting a mixed meal. The test is considered positive if the patient develops neuroglycopaenic symptoms in the presence of a capillary plasma glucose level of 3.0 mmol/L or less [19,33]. As previously detailed, venous blood may give false positive results since postprandial glucose concentrations in venous samples may be 2.5 mmol/L lower than in corresponding capillary samples [10].

The prolonged (5 h) 75 g glucose tolerance test is not recommended in the investigation of hypoglycaemia as many healthy subjects will have a false positive result, especially if venous blood samples are collected [30,34,35,36].

**Table 1 diagnostics-13-03275-t001:** Causes of spontaneous hypoglycaemia in adults.

HYPERINSULINEMIC HYPOGLYCAEMIA	**Endogenous hyperinsulinaemia:** Insulinoma, upper gastrointestinal tract surgery including bariatric surgery. **Non-insulinoma pancreatogenous hypoglycemia syndrome** (NIPHS) **Autoimmune insulin syndrome** **Factitious hypoglycaemia:** Administration of insulin, sulfonylurea
HYPOINSULINEMIC HYPOGLYCAEMIA	**Alcohol** **Drugs** **Critical illnesses:** Hepatic, renal, or cardiac failure, sepsis, malaria **Endocrinopathy:** Hypoadrenalism, hypopituitarism **Non-islet cell tumour hypoglycaemia** (NITCH): Mesenchymal, epithelial and hematopoietic tumours **Inherited metabolic disorders**: Glycogen storage disorders, disorders of gluconeogenesis, mitochondrial diseases, fatty acid oxidation disorders, hereditary fructose intolerance

**Figure 1 diagnostics-13-03275-f001:**
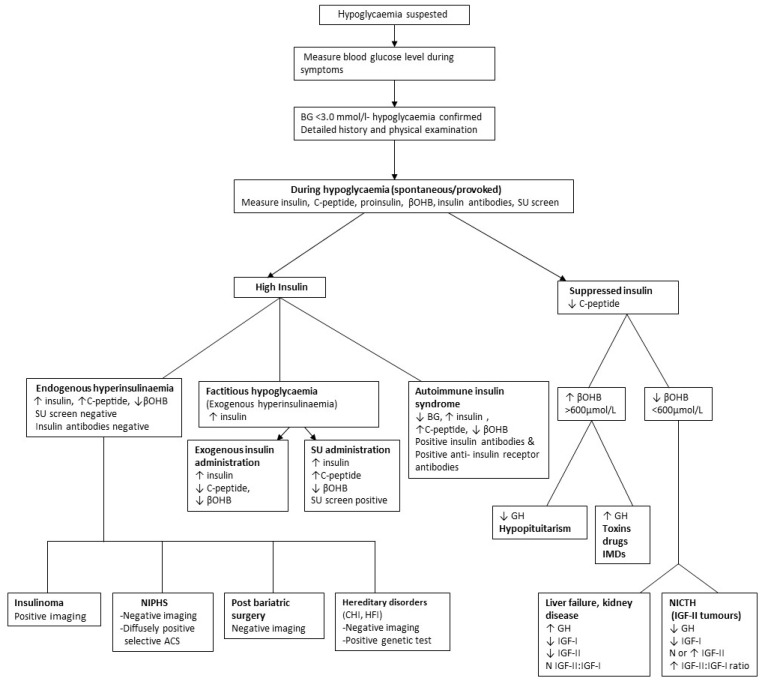
Algorithm for identifying cause of spontaneous hypoglycaemia. **Abbreviations**: BG: blood glucose; βOHB: β-hydroxybutyrate; SU: sulfonylurea; GH: growth hormone; NIPHS: Non-insulinoma pancreatogenous hypoglycaemia syndrome; CHI: congenital hyperinsulinism; HFI: hereditary fructose intolerance; selective ACS: selective arterial calcium stimulation; IGF: insulin-like growth factor, ↑: high; ↓: low.

**Table 2 diagnostics-13-03275-t002:** List of medications reported to be associated with hypoglycaemia in the non-diabetic patient.

Drug Type	Drug
Antimicrobials	Fluoroquinolones—clinafloxacin Pentamidine Quinolones—gatifloxacin, ciprofloxacin, moxifloxacin, levofloxacin Sulfamethoxazole
Anti-malarials	Artesunate Quinine Quinidine
Antipsychotics	Lithium Valproate—in neonatal exposure
Analgesic	Salicylates—especially in children
Cardiac agents	Beta blockers Angiotensin-converting enzyme inhibitors Disopyramide Cibenzoline Indomethacin
Other	Ethanol Insulin Growth Factor 1 Mifepristone—in labour Somatostatin analogues

## 5. Differential Diagnosis of Documented Hypoglycaemia

An algorithm is provided in Figure 1, which will elucidate most causes of postprandial (alimentary) and fasting (post-absorptive) hypoglycaemia.

Patients with hypoglycaemia are classified into those with (inappropriate) hyperinsulinaemia or (appropriate) hypoinsulinaemia (Table 1). The characteristic of hyperinsulinaemic hypoglycaemia is measurable, not necessarily high, insulin concentrations which are inappropriately high in the presence of hypoglycaemia [37,38,39].

Hypoglycaemia may further be sub-classified as ketotic or non-ketotic, usually by measuring plasma β-OHB. Low β-OHB during hypoglycaemia is indicative of hyperinsulinaemia or increased insulin-like activity (insulin-like growth factors—IGFs) since these suppress lipolysis and ketogenesis [29]. Low β-OHB may also occur in hypoinsulinaemic hypoglycaemia due to liver failure (site of ketone synthesis), anorexia nervosa or starvation (absence of fat and lipolysis) and inherited metabolic disorders (IMDs). Other causes of hypoinsulinaemic hypoglycaemia are associated with moderate to marked ketonaemia.

### 5.1. Postprandial (Alimentary) Hypoglycaemia

Postprandial (alimentary) hypoglycaemia is due to inappropriate endogenous hyperinsulinemia and generally occurs two to four hours after a meal. It is relatively common after bariatric surgery or other major gastric surgery and is then known as the late dumping syndrome [39,40]. Following surgery, the rapid transition of food to the small intestine causes an early and higher glycaemic peak and increased glucagon-like peptide 1 secretion, triggering excessive insulin secretion and subsequent postprandial hypoglycaemia [41,42].

Otherwise, postprandial hypoglycaemia is rare but may be due to autoimmune insulin syndrome (AIS), alcohol, non-insulinoma pancreatogenous hypoglycaemia syndrome (NIPHS) or hereditary fructose intolerance, or it is idiopathic [42].

AIS should be considered in patients with autoimmune disease who present with postprandial hypoglycaemia, inappropriately high insulin, incompletely suppressed C-peptide levels, a high insulin/c-peptide molar ratio > 1, high insulin autoantibody titres (without previous exposure to insulin) and the presence of macroinsulin [43,44,45,46,47]. In AIS, insulin released post-prandially binds to the insulin antibody resulting in hyperglycaemia, and then as insulin is released from the insulin antibody, hypoglycaemia ensues. Occasionally, the postprandial hypoglycaemia may be delayed and appears to present with fasting hypoglycaemia. Similar mechanisms have been proposed for the extremely rare occurrence of postprandial hypoglycaemia due to insulin-binding paraproteins [48].

NIPHS is characterised by diffuse nesidioblastosis and endogenous postprandial hyperinsulinaemic hypoglycaemia, typically but not invariably, three to four hours after meals [49,50]. Imaging studies are negative for insulinoma, and the diagnosis is confirmed by selective arterial pancreatic calcium stimulation, causing a diffuse hyperinsulinaemic response from multiple pancreatic vascular territories.

Milder forms of inherited metabolic disorders (IMDs) may rarely present with postprandial hypoglycaemia in adults and include hereditary fructose intolerance (HFI), congenital endogenous hyperinsulinemia and three subtypes of congenital disorders of glycosylation (CDG) other than type 1 [51,52]. HFI presents with abdominal symptoms with or without postprandial hypoglycaemia following fructose or sucrose ingestion. In CDG, hypoglycaemia is rarely the presenting feature and never occurs in isolation but rather as part of a clinical syndrome, which should prompt further investigation [52].

All causes of fasting hypoglycaemia may be associated and very rarely present with postprandial hypoglycaemia. Idiopathic postprandial (reactive) hypoglycaemia may, therefore, only be confidently diagnosed after the exclusion of fasting hypoglycaemia.

Postprandial hypoglycaemic symptoms without fulfilling Whipple’s triad, previously termed “reactive hypoglycaemia”, is a functional disorder now known as a postprandial syndrome, in which symptoms are not due to hypoglycaemia [53].

### 5.2. Fasting (Non-Prandial) Hypoglycaemia

#### 5.2.1. Hyperinsulinaemic Hypoglycaemia

Although rare, the commonest cause of endogenous hyperinsulinaemic hypoglycaemia is insulinoma, which is characterised by inappropriately high insulin and/or proinsulin, high C-peptide and suppressed or low BOHB in the serum. Pure proinsulinoma may be missed if very specific insulin assays [54,55] are used, and measurement of proinsulin is advised [1].

In proven insulinomas, genetic studies for Multiple Endocrine Neoplasia (MEN) should be undertaken since insulinoma is the commonest pancreatic endocrine manifestation of Type 1 MEN [56]. Selective pancreatic arterial calcium stimulation, endoscopic ultrasound and intra-operative ultrasound may be of value in the localisation of insulinoma; other imaging techniques are unreliable and may be misleading.

Other causes of hyperinsulinism, including factitious hypoglycaemia due to surreptitious ingestion of oral hypoglycaemics, autoimmune hypoglycaemia and reactive hypoglycaemia, must be excluded before making a diagnosis of insulinoma. This is especially important for accidental or factitious insulin secretagogue-induced (e.g., sulphonylureas) hypoglycaemia, which may produce an identical clinical and biochemical picture to insulinoma. Exclusion is essential by showing an absence of these drugs in blood or urine at the time of hypoglycaemia, and may prevent unnecessary laparotomy [57]. Type B insulin resistance, a very rare condition due to insulin receptor antibodies (IR-A), usually presents with hyperglycaemia but may also uncommonly present with fasting or postprandial hypoglycaemia and similar biochemical pictures to insulinoma [58,59]. These patients, however, will have one or all of hyperadiponectinaemia, hypotriglyceridaemia, acanthosis nigricans and autoimmune disease. The diagnosis of IR-A-mediated hypoglycaemia requires a demonstration of IR-A in the serum [47].

Hypoglycaemia due to exogenous insulin administration is easily distinguished from endogenous hyperinsulinaemia by inappropriately high insulin levels in the presence of low or suppressed C-peptide levels. Specific insulin assays, however, may fail to detect synthetic insulins and misclassify exogenous hyperinsulinaemic hypoglycaemia as hypoinsulinaemic hypoglycaemia, leading to unnecessary and misleading investigations [60,61]. Each laboratory should therefore be aware of the limitations of their insulin assays, and if required, samples should be referred to the specialist laboratory using a quantitative mass spectrometry-based method to exclude factitious hypoglycaemia due to exogenous insulin administration.

#### 5.2.2. Hypoinsulinaemic Hypoglycaemia

Hypoinsulinaemic (suppressed insulin and c-peptide) hypoglycaemia typically occurs spontaneously or may be provoked by fasting. Causes include tumours, endocrine hormonal deficiencies, organ failure, malnutrition including starvation and IMD (Table 1 and Figure 1).

Non-islet cell tumour hypoglycaemia (NICTH) is defined as tumour-induced hypoglycaemia not due to insulinoma. NICTH is most commonly due to excessive tumour secretion of abnormal insulin-like growth factor-II (big IGF-11) but also includes other very rare causes of hypoglycaemia due to insulin-binding paraproteins or tissue destruction by tumour causing major organ failure or endocrine disease [48].

In NICTH, the insulin-like activity of big IGF-II leads to hypoglycaemia by reducing glucose hepatic glucose output and increasing peripheral glucose uptake and also inhibits lipolysis and ketogenesis. Hypoglycaemia suppresses pancreatic B cell secretion [58]. Feedback of big IGF-II on the hypothalamic–pituitary axis inhibits growth hormone (GH) secretion with subsequent lowering of GH-dependent IGF-I and IGF-binding proteins secreted by the liver. Tumours secreting big IGF-II are characterised by an increased total IGF-II:IGF-I ratio, hypoglycaemia with suppressed insulin and C-peptide and inappropriately low GH and β-OHB levels [62,63].

Hypoglycaemia is an uncommon presentation of cortisol deficiency (hypoadrenalism) in adults without diabetes. It may be more severe in those with secondary (hypothalamic–pituitary failure) hypoadrenalism due to concomitant loss of growth hormone, an insulin antagonist. Primary hypoadrenalism (Addison’s Disease) is associated with Type 1 diabetes and may present in these patients with new-onset hypoglycaemia and decreasing insulin requirements, particularly in long-standing diabetes due to loss of the counter-regulatory hormones, glucagon and noradrenaline [64]. A single measurement of low serum cortisol in a hypoglycaemic sample is insufficient evidence of hypoadrenalism. Low serum cortisol may be due to the adaptation of the counter-regulatory mechanism to recurrent hypoglycaemia and may erroneously suggest hypoadrenalism [19,65]. Appropriate stimulation tests for confirmation of adrenal failure are needed to avoid misdiagnosis due to false positive results. If hypoadrenalism is confirmed, an adrenocorticotrophin (ACTH) measurement will distinguish primary (high ACTH) from secondary (inappropriately low ACTH) hypoadrenalism and direct further investigation of the underlying disease process [20].

Fasting hypoinsulinaemic hypoglycaemia due to IMD may be ketotic (glycogen storage disorders other than type 1 and disorders of gluconeogenesis) or non-ketotic (fatty acid oxidation disorder or glycogen storage disorder type I). Further detailed investigation of these disorders is provided elsewhere [51].

Spontaneous hypoinsulinaemic hypoglycaemia may occur in patients with known severe comorbidity, such as advanced liver disease (glycogen depletion and reduced gluconeogenesis), renal failure (reduced gluconeogenesis and reduced insulin clearance), malnutrition and anorexia nervosa (glycogen depletion and reduced gluconeogenesis due to substrate deficiency) [29]. Awareness and prevention are important and further investigation is unnecessary unless hypoglycaemia from another cause is suspected.

## 6. Analytical Considerations: Insulin, C-Peptide, Proinsulin and Their Antibodies

Specific insulin assays have replaced non-specific insulin assays (previously termed immunoreactive insulin), which measure not only insulin but also detect proinsulin and its fragments. Unlike immunoreactive insulin assays, specific insulin assays may fail to detect synthetic insulins and proinsulin. In consequence, they may fail to detect factitious, felonious and accidental exogenous insulin administration and insulinomas exclusively secreting proinsulin [60,61].

Proinsulin normally represents less than 10% of circulating immunoreactive insulin [66]. The most important use of the proinsulin assay is in the diagnosis of an insulinoma secreting exclusively proinsulin (proinsulinoma).

C-peptide is co-secreted with insulin from the pancreas in equimolar concentrations. Its major clinical use is in the detection of exogenous insulin-induced hypoglycaemia. The c-peptide/insulin molar is >1 except in exogenous insulin administration and AIS. C-peptide is cleared by the kidneys and is, therefore, elevated in renal impairment. This may pose problems in investigating hypoglycaemia in patients with renal disease [67].

Insulin, proinsulin and C-peptide immunoassays are potentially subject to interference from non-analyte antibody-binding substances, including antibodies to insulin and proinsulin. It is, therefore, essential that laboratories investigating hypoglycaemia offer, as a minimum, the measurement of insulin, C-peptide and proinsulin as an inconsistency in the results could point to immunoassay interference in an assay.

Anti-insulin antibodies (IA) can be raised in response to exogenous insulins, but this is less common with human insulins than previously administered animal insulins. They are usually not clinically significant and are of low affinity [47] but cause immunoassay interference.

Insulin autoantibodies (IAA) occur in patients never exposed to exogenous insulin. They may be present in high titres and be of high affinity and, thus, have the potential to be clinically significant. Whilst IAA are mostly clinically insignificant, they may cause AIS. IAA, which is considered a sine qua non for the diagnosis of AIS, may, however, also be present in non-hypoglycaemic individuals and even rarely in patients with insulinoma.

In summary, IA and IAA may be of no immunoassay, clinico-pathological significance; be of no pathological significance but interfere in laboratory immunoassays, giving rise to erroneous laboratory results and clinical confusion; or be of pathological significance by causing autoimmune hypoglycaemia.

## 7. Pitfalls to Avoid

Failure to recognise sub-acute neuroglycopaenic symptoms as a clinical manifestation of spontaneous hypoglycaemia, leading to delayed or missed diagnosis.Failure to use appropriate cut-offs dependent on sample type (serum or whole blood) for defining biochemical hypoglycaemia, erroneously confirming or excluding hypoglycaemia.Failure to recognise spurious hypoglycaemia and pseudohypoglycaemia.Failure to confirm hypoglycaemia by not documenting Whipple’s triad.Failure to recognise the limitations of blood glucose meters and continuous glucose monitoring (CGM) in the identification of biochemical hypoglycaemia.Inappropriate use of obsolete investigations, such as the prolonged oral glucose tolerance test.Failure to follow protocols for the prolonged fast, particularly in terminating the fast prematurely before laboratory-confirmed hypoglycaemia and failure to test for (subtle) neuroglycopenia.Failure to recognise that a few healthy individuals may have plasma glucose concentrations in the range of 3.0 mmol/L or less following prolonged fasting.Failure to provide hypoglycaemic samples for measurement of pancreatic hormones, counter-regulatory hormones and non-glucose substrates.Measurement of pancreatic hormones, counter-regulatory hormones and non-glucose substrates in non-hypoglycaemic samples.Failure to recognise assay limitations, in particular very specific insulin immunoassays failing to detect pure proinsulinomas and exogenous insulin abuse.Failure to exclude factitious and accidental hypoglycaemia, AIS and NIPHS before diagnosing insulinoma.Failure to recognise that insulinoma may very rarely present with hypoinsulinaemia and ketosis (c-peptide and proinsulin always inappropriately raised).Failure to measure insulin antibodies in a hypoglycaemic sample with high insulin levels and mislabelling it as factitious hypoglycaemia.Failure to recognise that IA and IAA may be of no immunoassay and clinico-pathological significance or be of no clinico-pathological significance but interfere in laboratory immunoassays, giving rise to erroneous laboratory results and clinical confusion.Failure to exclude causes of fasting hypoglycaemia before diagnosing idiopathic postprandial (alimentary) hypoglycaemia.Failure to recognise that low serum cortisol during hypoglycaemia may erroneously indicate hypoadrenalism, hence the need for appropriate stimulation tests for confirmation of endocrinopathy.

These pitfalls could mislabel healthy individuals as “hypoglycaemic”, resulting in the “worried well syndrome”; exposing the patient to unnecessary evaluation, expense and potential harm or missing treatable and preventable causes of spontaneous hypoglycaemia.

## 8. Conclusions

The investigation of suspected spontaneous hypoglycaemia involves a high index of clinical suspicion. It is essential to confirm hypoglycaemia with Whipple’s triad, and if confirmed, the underlying cause may be determined using appropriate and targeted investigation.

There are, however, many pitfalls for the unwary, and if avoided, clinical acumen and appropriate investigations will almost always uncover the cause of spontaneous hypoglycaemia so that optimal patient management may be implemented.

## Data Availability

An electronic literature search in PubMed was performed using words spontaneous nondiabetic hypoglycaemia, postprandial hypoglycaemia, reactive hypoglycaemia, hyperinsulinemic hypoglycaemia.

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
