# Peer review of "Investigation and Causes of Spontaneous (Non-Diabetic) Hypoglycaemia in Adults: Pitfalls to Avoid"

_diagnostics, 2023, doi:10.3390/diagnostics13203275_

Round 1

Reviewer 1 Report (New Reviewer)

This is a clear, well-written paper that presents information about the research and causes of spontaneous (non-diabetic) hypoglycemia in adults. The years of the majority of publications shown in the references section indicate that this issue is a multi-decade challenge in diagnosing spontaneous hypoglycemia. Regardless, the main advantage of this review is the pitfalls to avoid during diagnostics, since they may mislabel healthy individuals as “hypoglycemic” or misdiagnose treatable or preventable causes of spontaneous hypoglycemia. Thus, this paper can be attractive for investigators and clinicians focused on preventative or curative anti-hypoglycemic therapy.

Minor point:

1.      Page 3, subtitle 3: Investigation instead invrstigation

2.      References: use Page Layout in formatting references

Reviewer 2 Report (New Reviewer)

This article is very informative and will be helpful for the clinical laboratory and clinicians. 

This manuscript is a resubmission of an earlier submission. The following is a list of the peer review reports and author responses from that submission.

Round 1

Reviewer 1 Report

This is overall a very superficial article and not properly updated.

Section 1. Introduction is not updated, with the vast majority of references to articles published more than 20 years ago.

Section 2. Tests for Provoking Hypoglycaemia is not properly detailed.  The description of the different tests is quite superficial.

Section 4. Analytical considerations: Insulin, C-peptide, Proinsulin and their antibodies is again very superficial, including just a short statement for each individual biochemical parameter.

References are too old and not appropriate. Among the 65 references included by the authors, only 5 articles were published in the last three years, while the vast majority of the articles included in the references have been published more than 20 years ago. The authors have even included articles published in 1938, 1972, 1974 and so on.

English grammar should be also significantly improved. This article needs to be extensively revised by a native English expert.

English grammar should be also significantly improved. This article needs to be extensively revised by a native English expert.

Reviewer 2 Report

The entirely review seems to be very well documented and organised.

However, I will feel the need to add a section about COVID-19 influence. The fact that COVID-19 induces transient hyperglycaemia, or sometimes even leads to newly onset diabetes, is an aspect increasingly attested on many recent studies. This infection seems to unbalance the glycaemic equilibrium. Maybe it is worth to be mentioned its impact in each of the section described in the present paper, or by adding a completely separate paragraph about what is known regarding the impact of COVID-19 in regulating glucose homeostasis, especially of inducing hypoglycaemia.

Also, a very important aspect is the role of hypoglycaemia in pre-diabetes state, which is reversible state, a non-diabetic condition.

The authors may consider these aspects and decide if worth to be mentioned in their paper as they conceived it.